# Structural dynamics of basaltic melt at mantle conditions with implications for magma oceans and superplumes

Arnab Majumdar[1,5], Min Wu[1,2,5], Yuanming Pan[3], Toshiaki Iitaka [4] & John S. Tse [1✉]

Transport properties like diffusivity and viscosity of melts dictated the evolution of the Earth's early magma oceans. We report the structure, density, diffusivity, electrical conductivity and viscosity of a model basaltic ($Ca_{11}Mg_7Al_8Si_{22}O_{74}$) melt from first-principles molecular dynamics calculations at temperatures of 2200 K (0 to 82 GPa) and 3000 K (40–70 GPa). A key finding is that, although the density and coordination numbers around Si and Al increase with pressure, the Si–O and Al–O bonds become more ionic and weaker. The temporal atomic interactions at high pressure are fluxional and fragile, making the atoms more mobile and reversing the trend in transport properties at pressures near 50 GPa. The reversed melt viscosity under lower mantle conditions allows new constraints on the timescales of the early Earth's magma oceans and also provides the first tantalizing explanation for the horizontal deflections of superplumes at ~1000 km below the Earth's surface.

[1] Department of Physics and Engineering Physics, University of Saskatchewan, Saskatoon, SK S7N 5E2, Canada. [2] College of Materials Science and Engineering, Zhejiang University of Technology, Hangzhou 310014, PR China. [3] Department of Geological Sciences, University of Saskatchewan, Saskatoon, SK S7N 5E2, Canada. [4] Discrete Event Simulation Research Team, RIKEN Center for Computational Science (R-CCS), 2-1 Hirosawa, Wako, Saitama 351-0198, Japan. [5] These authors contributed equally: Arnab Majumdar, Min Wu. ✉email: john.tse@usask.ca

Basaltic melts are the most common magmas produced from partial melting of the mantle and play key roles in diverse geological processes from the core and crust formation to mantle convection, plate tectonics and surficial volcanic activity[1–4]. Similarly, basaltic melts and associated rocks have long been known to represent the most important components in the lunar and Martian crusts as well as some meteorites[1–5]. Average basalt contains 49.97 wt.% $SiO_2$, 15.99 wt.% $Al_2O_3$, 9.62 wt.% CaO, 6.84 wt.% MgO and 7.24 wt.% FeO as the major components[6] as well as various alkali cations (Li$^+$, Na$^+$, and K$^+$), water and other minor constituents. However, experimental determination of structures and physical properties of basaltic melts under most mantle conditions, especially those corresponding to the lower mantle, is challenging[2]. In spite of the present-day mantle being mostly solid, several studies have suggested that the mantle at the early stages of the Earth's history was mostly molten[7,8]. Moreover, the present-day low velocity zone (LVZ) with decreased seismic velocity but increased electrical conductivity in the upper mantle has been attributed to small degrees of partial melting, dominantly basaltic in composition. Also, the present-day core–mantle boundary (~135 GPa, 4000 K and 2900 km below the Earth's surface) has been proposed to contain partial melts, presumably dominantly basaltic in composition as sources for superplumes[1,8–10].

Despite a deep interest in understanding the structures and various properties of basaltic melts under mantle conditions and corresponding glasses under mantle conditions and numerous previous studies, both experimental[11–14] and theoretical[15–17], being available in the literature, a clear and unambiguous explanation of the structural and density changes is still awaited. In all previous theoretical and experimental studies, both silicon and aluminum with respect to oxygen atoms have been reported to transform from a dominantly tetrahedral structure at low pressure to an octahedral configuration at high pressure. So far, the exact pressure and temperature are not known for the onset of the structural changes, and also questions remain whether fourfold and sixfold coordinated Al and Si atoms can exist simultaneously. Another intriguing hypothesis is the co-existence of fivefold coordination as an intermediate state along with the tetrahedral and octahedral structures[18]. More importantly, the presence of fivefold coordinated Si and Al atoms has been suggested to have a significant effect on diffusivity and viscosity of melts. Unfortunately, available measurements provide limited information on these issues[2,4], because most experiments are limited to pressures <15 GPa due to technical difficulties. Likewise, theoretical calculations on transport properties such as viscosity of basaltic melts under deep mantle conditions are restricted to simple model systems such as $MgSiO_3$ and $CaAl_2Si_2O_8$[19–21] Therefore, quantitative data on the effects of Si, Al, and other major cations on the structures, densities and transport properties of basaltic melts as a function of pressure are essential.

In this work, we have investigated the structures, densities, and transport properties of a more realistic model basaltic melt consisting of CaO, MgO, $Al_2O_3$, and $SiO_2$ at 2200 (0–82 GPa) and 3000 K (40–70 GPa) by first-principle molecular dynamics (MD) calculations. The objective of this study is to relate the diffusivity, electrical conductivity, and viscosity, with local structural changes. In particular, anomalies in the reversal of the transport properties were predicted under the lower mantle conditions and have been attributed to temporal atomic interactions at high pressure which are fluxional and fragile, and have important implications for the mantle's electrical conductivity profile, the timescales of the early Earth's magma oceans, and the origin and upwelling of superplumes from the core–mantle boundary. We show that the reversed melt viscosity under lower mantle conditions near 50 GPa not only provides support for short timescales of magma oceans at a few million years but also provides plausible explanation for the horizontal deflections of superplumes at ~1000 km below the Earth's surface.

## Results and discussion

**Structures and structural transformations.** The structural evolutions with increasing pressure, as indicated by the equation of states and the relevant atomic radial distribution functions (RDFs), are in substantial agreement with previous reports. The results are summarized and compared with earlier works in Supplementary Figs. 1–3. Here, the discussion is focussed on how the local structural changes affect the transport properties. At low pressure, the Si–O coordination increases gradually from fourfold (tetrahedra; at zero pressure) to sixfold (octahedra; Fig. 1a). The average coordination number (CN) is often the quantity reported in experiments and most theoretical studies. Here, the evolution of the relative amount of different Si–O coordination with pressure is reported in Fig. 1a. The basaltic melt, like most other silicate melts, has fourfold Si–O coordination at ambient pressure. At the onset of densification at 18 GPa (see EOS of Supplementary Fig. 1), the melt is composed of mixed fourfold (66%) and fivefold (34%) coordinated Si. The melt above 38 GPa is comprised of a mixture of fourfold (<5%), fivefold (25%), and sixfold (70%) Si coordination. Above 50 GPa, the transformation from fourfold to sixfold Si coordination is complete. The variation of the average Si–O CN with pressure is compared to the results extracted from experiments and calculations[2,22–24] in Fig. 1b. The increment from fourfold to sixfold coordination proceeds in two distinct regions. Initially, the Si–O coordination increases rapidly with pressure but slows down above 23 GPa. Inspection of structures obtained from the simulations in this pressure region shows the closing of the Si–O–Si bridging angle of two connected tetrahedra. This configuration facilitates the compression of a fifth oxygen atom into the first coordination shell forming edge shared pentahedra and tetrahedra. On further compression, edge-sharing polyhedra become more abundant and concomitant with increase in the Si–O CN. At 20 GPa, the O–Si–O angles widen and the average Si–O distance increases to accommodate the fifth, nearest neighbor oxygen atom, which can be verified from the bond angle distribution shown in Supplementary Fig. 4a. The ~90° is the octahedral O–Si–O and the ~170° is the axial (linear) O–Si–O showing that the local environment is almost octahedral. Moreover, the Si–O interaction at 20 GPa becomes more ionic as the fourfold coordination is gradually lost. This trend continues with a further increase in Si–O coordination at higher pressures. Snapshots of selected structural configurations with respect to Si–O bonding under pressure are illustrated in Supplementary Fig. 5.

Changes in the Al–O CN with pressure can be explained in a similar manner (Fig. 1c, d). At ambient pressure, the Al coordination is mainly fourfold. By 18 GPa, the fourfold coordination decreases drastically (31%) and gives rise to the intermediate fivefold coordination (69%). A similar trend was observed in aluminosilicate liquids[3]. In comparison with Si, the increase for the Al coordination starts at a much lower pressure. The fraction of fourfold-coordinated Al atoms (Fig. 1c) drops sharply and becomes mostly fivefold coordinated at 18 GPa. Above 23 GPa, the prevalence of the octahedral coordination (13%) is clear. From the bond angle distribution of O–Al–O (Supplementary Fig. 4b), the O–Al–O becomes six coordinated much faster than O–Si–O. The fivefold-coordinated Al remains stable over a considerable range of pressure with an equal mixture of five- and six-coordinated Al atoms at 50 GPa and gradually changes to $AlO_6$. Above 70 GPa, a sevenfold coordination starts to appear and becomes dominant on further compression.

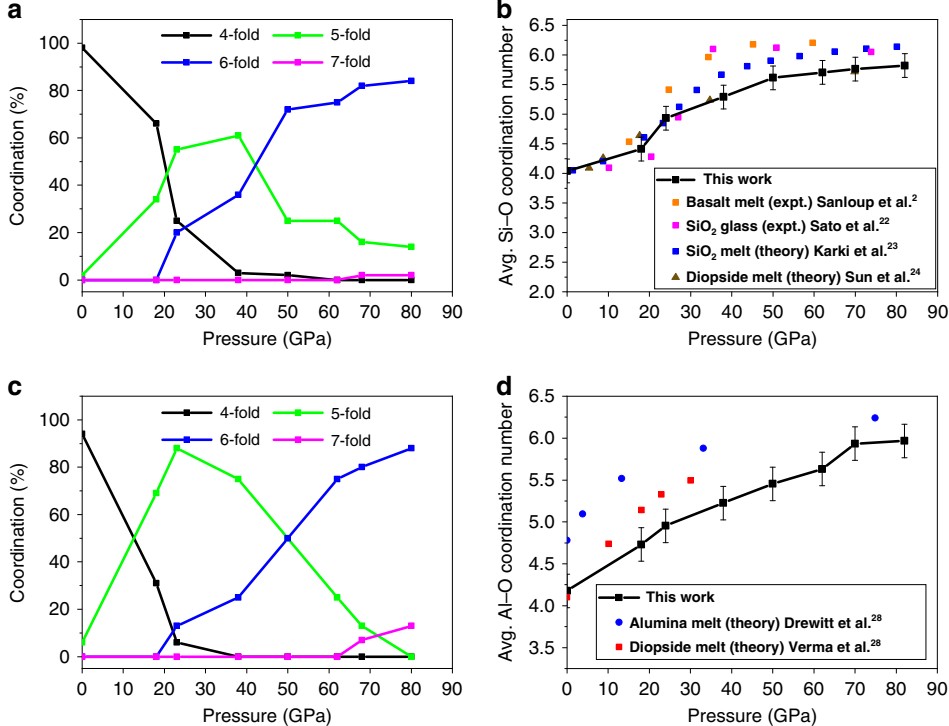

**Fig. 1 Coordination numbers of Si and Al as functions of pressure. a** Coordination percentage vs pressure of Si atoms with respect to O atoms at 2200 K, **b** average Si–O coordination number vs pressure and compared with related silicate glasses. **c** Coordination percentage vs pressure of Al atoms with respect to O atoms at 2200 K and **d** average coordination number of Al atoms vs pressure. The error bars represent the uncertainty of the calculated mean coordination numbers, which arise from uncertainties in determining the radial cut-off from the radial distribution function.

Snapshots of selected structural configurations to emphasize the Al–O bonds under pressure are illustrated in Supplementary Fig. 6.

Evidence for fractional changes in Si–O and Al–O coordinations in basaltic melts is not available directly from experiments, although averaged CNs have been reported several times for both melts and glasses[2,12,25,26]. It is, however, relevant and informative to compare the structural changes in the basaltic melt with other silicate melts and glasses. Under ambient pressure, the calculated average Si coordination in the basaltic melt of 4.04 (Fig. 1b), with a small fraction (~2%) of fivefold coordination is similar to that found in MgSiO₃ glass[27]. The calculated Al CN of 4.17 (Fig. 1d) at ambient pressure is similar to the experimental value of 4.1 obtained for liquid CaAl₂Si₂O₈[28] although it is slightly less than that for alumina[29]. The relative proportions of the CNs also compare well with anorthite liquid at 3000 K[30]. It should be cautioned that the comparison made here is only qualitative as there are differences in compositions, pressure, and temperatures of the basaltic melt and other systems. For the benefit of the ensuing discussion, we wish to point out that our simulations of the basaltic melt at 3000 K show a similar structural transformation sequence as that of 2200 K, except that, as expected, five- and six-fold coordinated Si–O and Al–O (Supplementary Fig. 7) appear at pressures lower than those at 2200 K.

So far, we only discussed the time average structure of the basaltic melt at 2200 K (and 3000 K). To explain the transport properties (*vide infra*), it is necessary to investigate the transient behavior of the Si–O and Al–O coordinations. We have calculated the temporal evolution of the Si–O and Al–O CN at selected pressures and summarize the results in Fig. 2. The dynamical variation of the CN provides a different viewpoint of the structure as anticipated from the average value. At 2200 K and up to 38 GPa, Si and Al atoms maintain their respective coordination over a long time (20 ps) and there is no significant bond breaking

under this pressure–temperature condition. Also, as described above, Si–O is a mixture of fivefold and sixfold coordinations; but the Al–O coordination is mostly fivefold. At 50 and 68 GPa and 2200 K, the temporal change in the local Al–O structure is much more prominent. Noting that in Fig. 2d, f, the time periods for the plot were reduced to 2 ps to emphasize the rapid formation and breaking of Al–O bonds within a very short time. This observation shows the instability and rapid inter-conversion of the Al–O polyhedra among fourfold, fivefold, sixfold, and even sevenfold coordinations, with significant impacts on the melt viscosity as shown later.

## Transport properties

**Diffusivity.** The diffusion coefficients (*D*) at 2200 K are summarized in Fig. 3a. The mean squared displacement from which diffusion coefficients are calculated is shown in Supplementary Fig. 8 for 0 GPa and 2200 K. The formula used is described in the "Methods" section and the data have been reported with an error bar of ±10%. The predicted diffusivity sequence of $D_{Mg} > D_{Ca} > D_{Al} \approx D_O > D_{Si}$ agrees with the observed order in a melt with the composition 20CaO–20Al₂O₃–60SiO₄ (wt.%) at 1 GPa and 1773 K[20]. The ratio $D_{Mg}/D_{Si} = 1.83$ is also consistent with literature data[31]. The diffusion rates are sensitive to local structural environments. The interactions between O and Mg (and Ca) atoms are relatively weak compared to the Si–O and Al–O bonds. Therefore, Mg and Ca move easily in the open space available at large volumes. For example, in liquid MgSiO₃, Mg was predicted to be the most mobile species[19]. The dependence of *D* on pressure at 2200 K and 3000 K are shown in Fig. 3a, b, respectively. As expected, the diffusion coefficients of all the species decrease with pressure. At 2200 K (Fig. 3a), there is a drop in the diffusion coefficients at ~23 GPa from roughly $2 \times 10^{-10}$ to $4 \times 10^{-11}$ m² s⁻¹ (average of all the ions). This change is attributed to the onset of the

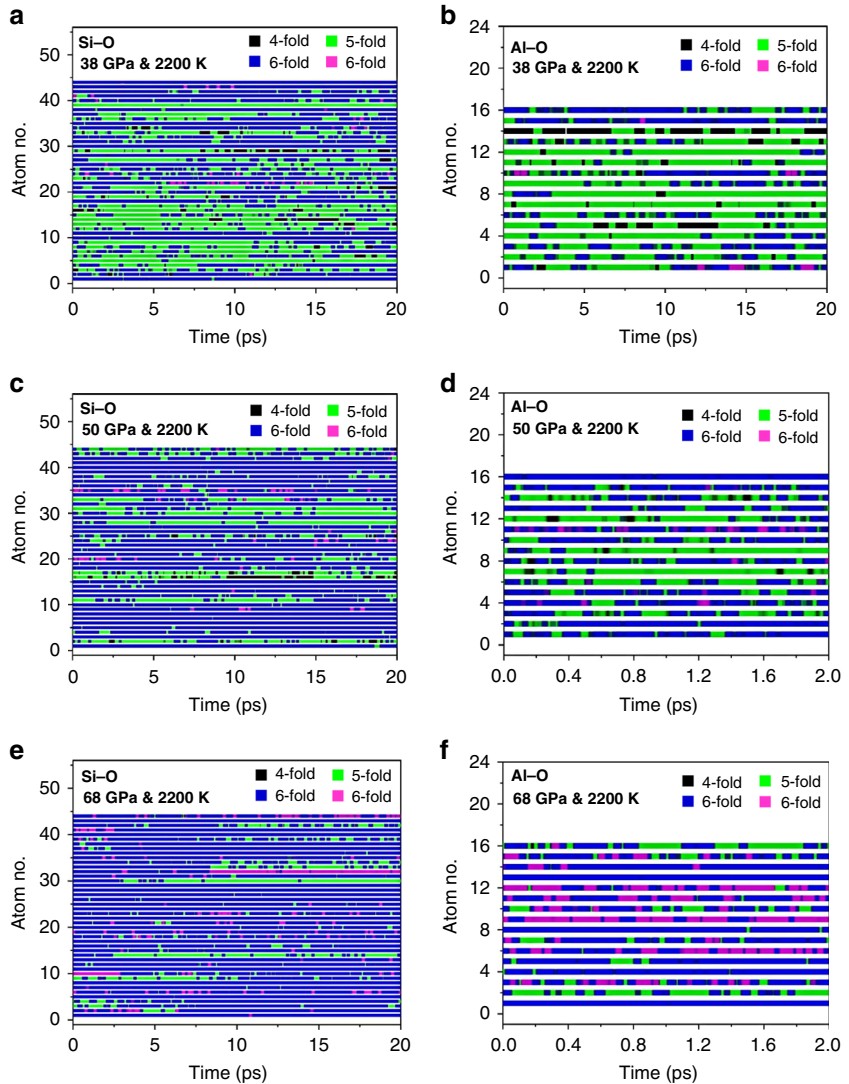

**Fig. 2 Temporal evolution of the coordination numbers.** For **a** Si at 38 GPa and 2200 K, **b** Al at 38 GPa and 2200 K, **c** Si at 50 GPa and 2200 K, **d** Al at 50 GPa, and 2200 K, **e** Si at 68 GPa and 2200 K, and **f** Al at 68 GPa and 2200 K.

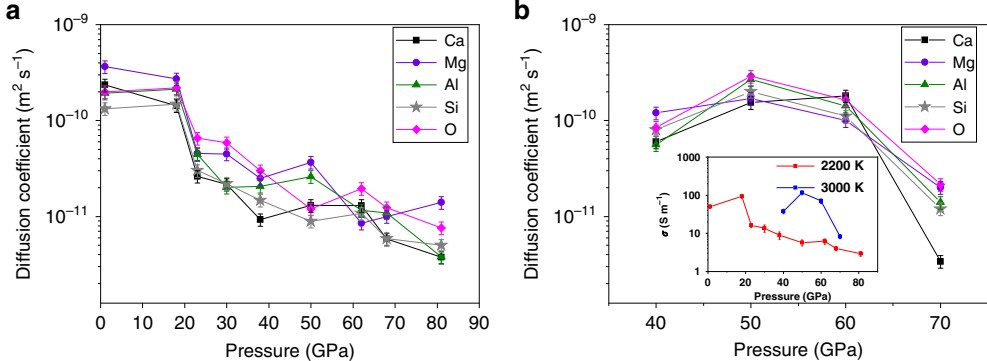

**Fig. 3 Diffusion coefficient of the individual species vs pressure.** Diffusion coefficients at **a** 2200 K and **b** 3000 K. (inset of **b**) Electrical conductivity at 2200 and 3000 K. The error bars are the uncertainties in the diffusion coefficients. The errors of the mean were determined by considering (i) different intervals of the time origin of the molecular dynamics steps and (ii) minor variation in slope of the straight line used to fit the mean squared displacement.

transformation from fourfold to higher coordination of both Si and Al. The diffusion rates continue to decrease with increasing pressure. Surprisingly, between 50 and 60 GPa, the diffusion coefficients of all the species increase before decreasing again. This reversal in the trend is not an artifact of the calculations and to confirm that,

we performed similar analysis in the same pressure range at 3000 K. Similar and even more pronounced anomaly is also predicted in this pressure range at 3000 K from our calculations (Fig. 3b). For purposes of clarity, in Fig. 3 we have presented only the data that we calculated. However, a comparison with the diffusion coefficients of

other related melts at 3000 K from previous studies can be found in Supplementary Fig. 9.

**Electrical conductivity.** From the Nernst–Einstein equation, the behavior of the electrical conductivities is expected to follow that of the diffusivities. Due to interatomic interactions, the ionic charge, $q_i$, in the melt is dependent on the local structure of the ion. The predicted conductivity of the basaltic melt at 2200 K and 0 GPa of *ca.* 50 S/m (Fig. 3b inset) is about ten times lower than that of $CaCO_3$ under similar conditions[32]. These theoretical results are consistent with the suggestion that the ionic conductivities of carbonatite melts are much higher than their silicate counterparts. The calculated electrical conductivities of the basaltic melt at 2200 K are comparable to 5 S/m measured on a natural sample of andesite melt at 1724 K[33]. The anomalously high conductivity of the model basaltic melt is due to the higher contents of the mobile $Ca^{2+}$ and $Mg^{2+}$ ions in comparison to real basalt. The melt electrical conductivity increases from ambient pressure to 20 GPa, then decreases and shows an anomalous maximum at 60 GPa and 50 GPa (Fig. 3b inset) for the cases of 2200 K and 3000 K, respectively.

**Viscosity.** While the evolution of the viscosity of basaltic melts with the temperature is rather well reported[34,35], its evolution with the pressure is yet to be fully understood, especially at very high pressure[36,37]. We must point out that in other recent theoretical works on determining the transport properties of melts, they have used classical MD[37–39]. This can impact results depending on the choice of empirical pair potentials used. On the contrary, our calculations are based on ab initio MD. In the works by Dufils et al.[37–39], the more realistic models of basalt, midocean ridge basalt (MORB) have been considered, which include more cationic species. These systems involve thousands of atoms for simulations that are beyond the scope of being treated using ab initio MD. Even then, our results are in the same order of magnitude. For example, they have concluded that for temperatures above 2273 K, even at 30–40 GPa, the viscosities are less than 100 mPa s, that can have implications on the geodynamics of the Archean mantle of the ancient Earth[37]. At 0 GPa and 2273 K, they calculated the coefficient of viscosity to be ~80 mPa s, in comparison with our corresponding magnitude of 50 mPa s at 0 GPa and 2200 K. Their magnitudes of well below 100 mPa s for 30 and 40 GPa and 3273 K are again similar to what we have obtained for these pressures and 3000 K. In another more recent work, Dufils et al. showed that for MORB at 1673 K and 0.5 GPa, the coefficient of viscosity is ~1200 mPa s[39]. This is large as the temperature is much lower and the higher viscosity is definitely expected and does not fall within the scope of our work. The Stokes–Einstein relation between particle diffusivity and fluid viscosity works well for comparatively simpler liquids such as metals[40,41]. However, there have been reports about discrepancies for fluids of more complex composition implying that the diffusion involves mechanisms that cannot be explained by such a simple model[42]. Our calculated viscosity reveals a slight drop from ambient pressure to 18 GPa at 2200 K (Fig. 4). The results seem to be at odd with an earlier report in which the predicted viscosities are much higher[20]. Apart from differences in the composition and number of atom of the basaltic melt models, we believe the discrepancy can be largely attributed to the use of a local density functional and the relatively short simulation time (<20 ps) used in the previous work. Here, we employed the entire converged MD trajectory (up to 150 ps) after thermal equilibrium to estimate the errors from averaging the viscosities computed at different time origins and with correlation lengths. Besides, the present results are consistent with other aluminosilicate melts

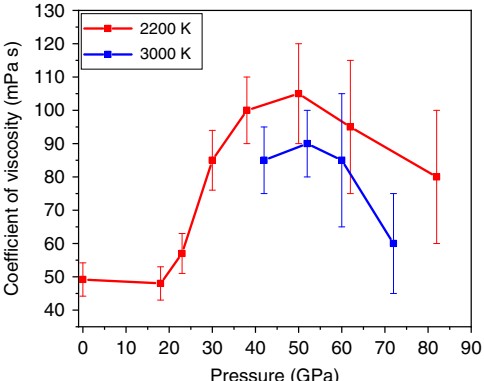

**Fig. 4 Coefficient of viscosity vs pressure.** Red and blue lines represent the data at 2200 K and 3000 K respectively. The error bars are the uncertainties in the coefficients of viscosity. The uncertainties of the mean values were determined by considering two possible sources of error in the calculations, that are (i) interval of the time origin of the molecular dynamics steps and (ii) oscillatory nature of the stress auto correlation function after it decays to zero, which is integrated to obtain the coefficient of viscosity.

that have been studied. For example, the diopside melt shows an initial decrease in viscosity with increasing pressure[43]. Between 0 and 18 GPa, both Si and Al are predominantly four-coordinated, and the fivefold coordination starts to appear above 20 GPa. The fivefold coordination has been proposed by Angell et al. to be a key ingredient for facilitating diffusion and viscous flow[18]. The explanation is as follows. A pure aluminosilicate liquid should consist of four coordinated Al and Si atoms, and all the oxygen atoms act as bridging oxygen (BO) atoms (corner-shared tetrahedra). Such systems should have very high viscosity owing to the high activation barrier. Upon pressure increase, the coordination of the aluminum atoms increases to five rapidly, weakening the Al–O bond, thus increasing the atom mobility and lowering the viscosity. Yarger et al.[44] suggested that the bond angle changes of the tetrahedral aluminosilicate network can be another contributing factor. The addition of alkali and alkaline earth elements acts as network modifiers that reduce the viscosity drastically[44]. Previous studies have shown that the viscous flow in aluminosilicate melts is dependent on the oxygen exchange between polymeric units[45]. The presence of non-bridging oxygen (NBO) atoms destroys the tetrahedral network structure resulting in a lower activation energy for oxygen exchange. The low activation energy enables high mobility, i.e., low viscosity. In this study, between 0 and 18 GPa, the decrease in the viscosity can be attributed to the rapid increase in the aluminum coordination with oxygen. Above 18 GPa, the viscosity increases steadily up to 50 GPa. Above 50 GPa, the coefficient of viscosity decreases for both 2200 and 3000 K, evident in Fig. 4, contrary to other studies on silicate melts[46]. We have already seen that, above 50 GPa, Al–O starts to appear in sevenfold coordination, thus weakening the Al–O bond further. Furthermore, on compression, the alkaline earth elements in our system, i.e., Ca–O and Mg–O, also attain higher CN (Supplementary Fig. 3), thus giving rise to even more NBO and lowering the viscosity further. On compression, both Si–O and Al–O polyhedra show increasing five and sevenfold coordinations as shown in Fig. 5 and Supplementary Figs. 5 and 6 that have been purported to be a contributing factor to increasing the NBO and thus increasing the ionic mobility. This is further supported by the temporal evolution of the CN (Fig. 2). Contrary to conventional expectation, the calculations reveal that, although the density (Supplementary Fig. 1b) and the Si–O and Al–O CN (Fig. 1b, d) all increase with pressure, the highly

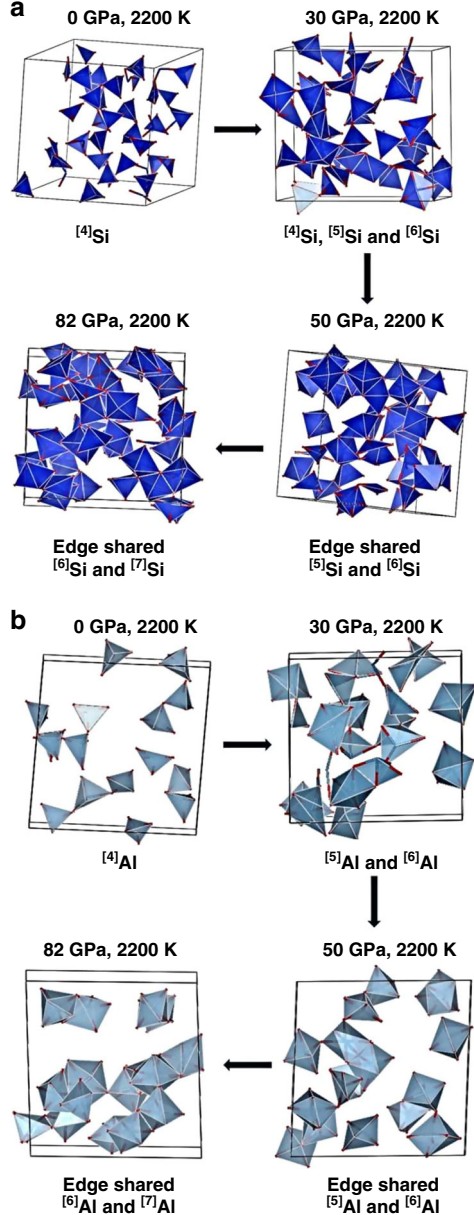

**Fig. 5 Si–O and Al–O polyhedra at different pressures. a** Depiction of Si-O linkages showing the transition from 0 to 82 GPa via 30 and 50 GPa at 2200 K and **b** polyhedra of Al-O linkages showing the transition from 0 to 82 GPa via 30 and 50 GPa at 2200 K. [4]Si, [5]Si, [6]Si, and [7]Si are the four, five, six, and sevenfold coordinated Si atoms. [4]Al, [5]Al, [6]Al and [7]Al are the four, five, six, and sevenfold coordinated Al atoms. The figures are not drawn to scale.

coordinated Si–O and Al–O bonds are more ionic and the temporal atomic interactions are fragile and fluxional (Fig. 2). The atoms at high pressure become mobile, thus reversing the trend in transport properties (Figs. 3 and 4). For purposes of clarity, in Fig. 4 we have presented only the coefficient of viscosity that we calculated. However, a comparison with that of other related melts reported in literature (both experimental and theoretical) can be found in Supplementary Fig. 10.

**Implications for superplumes and early Earth's magma oceans.** In comparison with previous calculations in simple systems such as $SiO_2$, $MgSiO_3$, and $CaAl_2Si_2O_9$[19–21], the multiple-component

system of $Ca_{11}Mg_7Al_9Si_{22}O_{74}$ investigated in the present study represents a significant step toward natural basaltic compositions but is still missing several components such as Fe and H. However, inclusion of these additional components, especially Fe, requires considerations of oxidation–reduction states and magnetic contributions with possible spin transitions, which are beyond the scope of this work using current computation capabilities. Our calculations for the more normal mantle temperature at 2200 K predict complete solidification of the model basaltic system above 82 GPa is consistent with the fact that the bulk of the lower mantle is in the solid state. Moreover, the theoretical results on transport properties of the model basaltic melt reported herein have important implications for the mantle's electrical conductivity profile, the time scales of the early Earth's magma oceans, and the origin and upwelling of superplumes from the core–mantle boundary. The electrical conductivity of 50 S/m for the model basaltic melt at ambient pressure and 2200 K (Fig. 3b inset) is in reasonable agreement with available experimental data for anhydrous silicate melts[47]. Remarkably, the maximum electrical conductivity of 139 S/m for the model basaltic melt at 2200 K and 18 GPa are similar to those measured for hydrous and hydrous carbonated basaltic melts[47] and are consistent with increased diffusion coefficients of Ca, Mg, and Al (Fig. 3b inset). Sifré et al.[47] showed that the presence of hydrous carbonated basaltic melts can account for both the seismic low velocities and the high electrical conductivities in the LVZ in the upper mantle. Moreover, the slight increase in electrical conductivity of the model basaltic melt at ~50–60 GPa may also contribute to the rapid rise in the electrical conductivity profile in the upper part of the lower mantle[48].

Various models such as the presence of significant partial melting[8,10], Fe-rich postperovskite[49], alloying with Fe-rich materials from the core[50] and slab-derived metallic melt[51] have been proposed to explain the origin of the ultra low velocity zone (ULVZ) at the core–mantle boundary. Labrosse et al.[52] suggested the ULVZ to be a remnant of an ancient basaltic magma ocean. Seismic tomography suggested that superplumes in the form of broad, quasi-vertical conduits are rooted at the core–mantle boundary and extend to ~1000 km below Earth's surface, where significant horizontal deflections occur[9]. The viscosity reversal of the model basaltic melt calculated at pressures of ~50 GPa (Fig. 4) provides the first tantalizing explanation for the horizontal deflection or stagnation of superplumes at the depth of ~1000 km[9]. If the predicted trend (Fig. 4) continues with depth and temperature, there may be implications for the ULVZ with melts of significantly lower viscosity than previously suggested.[37–39]

Magma oceans that formed from giant impacts during accretion are widely accepted to be responsible for the formation of the metallic core and the silicate mantle through differentiation as well as the atmosphere and hydrosphere through degassing[53]. Viscosity is an important parameter that controls virtually all the dynamic processes in early Earth's magma oceans. It is interesting to note that Karki and Stixrude[19] adopted a viscosity value of 48 (10) mPa s for anhydrous $MgSiO_3$ liquid at 70 GPa and 4000 K to obtain a Rayleigh number that lies in the regime of turbulent convection in magma oceans: i.e., cooling-induced crystal setting substantially influenced by the presence of turbulence. This viscosity is close to those calculated for the model basaltic melt at 70 GPa and 2200–3000 K (Fig. 4), whereas the calculated viscosity of anhydrous $MgSiO_3$ liquid at 70 GPa and 3000 K is an order of magnitude higher[19]. In addition, the timescales of magma ocean crystallization have been suggested to vary from thousands to millions of years, which depend largely on magma viscosity. Abe[54] assumed a melt viscosity of 100 Pa s to predict timescales of ~100–200 million years for magma oceans. More recent studies[55,56] using different assumptions have reduced the

timescales of magma oceans to a few million years. For example, such short timescales of magma ocean solidifications were obtained from radiative-convection equilibrium calculations with the assumed viscosity value of 0.1 Pa s for ultra-basic liquids at 3000 K[56]. However, previous calculations of anhydrous MgSiO₃ liquid at 3000 K[19] predicted this viscosity value to be valid only at pressures below ~40 GPa. On the other hand, our calculations with the reversed trend at ~50–82 GPa yield the viscosity values of ~0.1 Pa s for basaltic melts under most lower mantle conditions (Fig. 4), hence providing further support for the short timescales of magma oceans at a few million years[55,56].

## Methods

**First principles calculations**. Ab initio MD simulations have been performed on a model basaltic system. Constant volume and constant temperature (NVT) canonical ensemble was employed with appropriate Nosé thermostats[57]. The simulations were carried out using the Vienna ab initio Simulation Package program[58], expanding electron orbitals in the plane wave (PAW) basis set. PBE[59] functional was used keeping the kinetic energy cut-off of the plane wave 400 eV. The two-electron pseudopotential has been utilized for Mg instead of the *p*-valence one, as it compares very closely with *p*-valence pseudopotential[60] and also shown in Supplementary Fig. 11 for face centered cubic MgO, and reduces the computational expense significantly. The interatomic forces are computed for all the time steps from a fully self-consistent solution of the electronic structure to the Born-Oppenheimer surface, within the finite temperature formulation of density functional theory. The stoichiometry of the model basaltic material studied was a mixture of seven diopside (CaMgSi₂O₆) and four anorthite (CaAl₂Si₂O₈): i.e., Ca₁₁Mg₇Al₈Si₂₂O₇₄ containing 50.27 wt.% SiO₂, 15.51 wt.% Al₂O₃, 23.47 wt.% CaO, and 10.73 wt.% MgO. Our model basalt system contains more CaO and MgO in comparison to real basalt, as we have not considered the chemical effects of FeO, Li⁺, K⁺ and N⁺ in our simulations. A cubic supercell with a total of 244 atoms was constructed. Owing to the large size of the supercell and computational limitations, we ran all the simulations using just one k-point (Γ) to sample the Brillouin Zone. For the melt system, simulations were performed at 0, 18, 23, 30, 38, 50, 62, 68, and 82 GPa and 2200 K. To confirm the viscosity trend between 40 and 70 GPa, we also ran similar MD simulations at 40, 50, 60, and 70 GPa and 3000 K. At each pressure, the melt was first equilibrated using constant-pressure-constant temperature (NPT) ensemble MD at the desired temperature. A model cell was then determined from the average of the cell parameters from the NPT simulation of the equilibrated steps. In addition, NVT MD were then performed on this model. The time step used for the integration of the equation of motions was chosen to be 2.0 fs. All MD simulations were performed for 60,000 up to 120,000 time steps, equivalent to 0.12–0.24 ns. From monitoring the temporal evolution of the isotropic stress, temperature, and total energy, it was found that most systems equilibrated after ~10000 steps. The remaining atomic trajectory from the MD calculations was used in the analysis.

**RDF and CN**. Pair correlation function, also called RDF, g(r) gives an estimate as to how the density varies as a function of the distance from a reference particle. Thus, it represents how atoms are radially packed around each other. It is one of the quantities that can be directly compared with experimental data. The RDF is defined as,

$$g_{\alpha\beta}(r) = \frac{dn_{\alpha\beta}(r)}{4\pi r^2 dr \, \rho_\alpha}, \qquad (1)$$

where $\rho_\alpha = \frac{N_\alpha}{V}$ is the number density of the particles of type α. V is the volume of the system. In Eq. 1, $dn_{\alpha\beta}(r)$ is the number of β atoms around α atoms within a radial distance of r and r + dr. On integrating the RDF up to the first minimum gives the CN. The integration is performed according to the formula,

$$CN = 4\pi\rho \int_0^{r_m} g(r) r^2 dr, \qquad (2)$$

where $r_m$ is the first radial cutoff or the first minimum of RDF.

**Diffusion**. The self-diffusion coefficients $D_\alpha$ for elemental species α (Ca, Mg, Al, Si, and O) were calculated from the mean squared displacement (MSD) using the Einstein relation given as

$$D = \frac{1}{2N} \frac{d}{dt} \langle |\vec{r}(t) - \vec{r}(0)|^2 \rangle, \qquad (3)$$

where N = 1, 2, or 3.

**Electrical conductivity**. The motions of mobile ions in the melt result in weak electrical conductivity. The ionic conductivity is calculated from the

Nernst–Einstein equation,

$$\sigma = \frac{e^2}{K_B T} \sum_i n_i q_i^2 D_i, \qquad (4)$$

where σ is the electrical conductivity, $K_B$ the Boltzmann constant, T is the temperature, e is the elementary charge, $n_i$ is the number of *i*th ion species per unit volume, $q_i$ is the charge of the *i*th ion and $D_i$ is the self-diffusion coefficients of *i*th ion. We estimated the effective ionic charges from Bader analysis of the electronic charge distribution at several time steps[61]. An average value is then calculated for each ionic species.

**Viscosity**. The melt viscosity (η) was calculated from the stress-tensor auto-correlation function (SACF) using the Green-Kubo relation,

$$\eta = \frac{V}{3k_B T} \int_0^\infty \left\langle \sum_{i<j} \sigma_{ij}(t + t_0) \sigma_{ij}(t_0) dt \right\rangle, \qquad (5)$$

where $\sigma_{ij}$ (i, j = x, y, z) is the stress tensor computed at every MD step. The stress tensors of the model basaltic melt were checked to ensure that the pressure in the MD is isotropic. The decay of the SACF with time has been represented in Supplementary Fig. 12 for the cases of 0, 38, and 62 GPa at 2200 K. The convergence of the coefficient of viscosity with time has been shown in Supplementary Fig. 13 for the cases of 0, 38, and 62 GPa at 2200 K. The estimated errors were calculated using different time origins and time durations. Since accurate calculations of the viscosity are computationally very demanding, only a few pressure points were computed at 3000 K to confirm the anomalous increase of the viscosity predicted at 2200 K. The confidence levels were determined by considering two possible sources of error in the calculations, (i) interval of the time origin and (ii) oscillatory nature of the SACF after it decays to zero.

## Data availability
The data that support the findings of this study are available from the authors on reasonable request.

## Code availability
The codes used for analysis of the data of this study are available from the authors on reasonable request.

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

## Acknowledgements

This project was supported by the Natural Sciences and Engineering Research Council of Canada (NSERC). The authors would like to thank the University of Saskatchewan, WestGrid, and Compute Canada for providing with computing resources. MW thanks the support by the National Natural Science Foundation of China (Grant No. 51701180). The authors also wish to thank Dr. B.B. Karki for the initial coordinates of the model basalt system. This research is support by MEXT "Exploratort Chanllenge on Post-K Computer" (Cha;;enge of Basic Science - Exploring Exremes through Muli-Physics and Multi-scale Simulations).

## Author contributions

A.M, M.W and J.S.T carried out the computational simulations. A.M and M.W contributed equally to this whole work. A.M, M.W, Y.P, T.I and J.S.T contributed to the analysis of the results and writing of the manuscript.

## Competing interests

The authors declare no competing interests.
