## [Peer Review File · Nature Communications]

Reviewers' comments:

Reviewer #1 (Remarks to the Author):

This paper reports the properties of a molten basalt analogue up to 80 GPa by means of theoretical calculations. The major finding is that melt viscosity remains low up to the highest P investigated (80 GPa) with a non-monotonous P-evolution. Viscosity slightly decreases up to 20 GPa, then increases only to reach a maximum near 50 GPa (100 mPa.s) and then slowly decreases. These results differ from those obtained previously from theoretical calculations (Karki 2018 and refs therein) that did not report a slow decrease above 50 GPa, with viscosity values about 2-3 times higher than found here. Although experimental data on viscosity only exist up to modest P, experimental values are also 3-4 times higher than found here (i.e. below 10 GPa, cf highest T points in Sakamaki 2013). The decrease of viscosity above 50 GPa is attributed to the Si-O and Al-O bonds become ionic, explaining weakening of the melt structure and reduced viscosity. The most straightforward implication of such low viscosity is that the magma ocean must have cooled faster than previously thought. There are no molten basalts in the present day deep mantle except in ULVZs. Hence implications for the reported viscosity decrease in the solid lower mantle are quite stretched.

There are a number of mistakes in the literature and geological background (see below), however that can be corrected. I am more concerned by the mismatch between the present results on diffusivity and viscosity and previous theoretical and experimental results, while structural and density data are in general agreement with previous works.

Such mismatch, especially at modest P at which experiments can be done and are confirmed by other theoretical works, cast doubts on the new finding i.e. a decrease of viscosity at lower mantle conditions. For these reasons, I cannot recommend the paper for publication.

Geological background:

I do not think that it is appropriate to state climate change as being governed by the properties of magmas at depth. This induces the idea of a control from the deep Earth on climate change, which is inaccurate and dangerous for a scientist to state. Super-eruptions (e.g. traps) did influence climate in the past, but there is no such effect since CO₂ increase in the atmosphere from human-related activities.

Most implications related to the presence of basalts at depth are fine however at places the manuscript implies that there is partial melt in the lower mantle which is not the case except in ULVZs where T are high enough to induce partial melting. Mantle plumes may rise from low-shear wave velocity provinces (that do have some ULVZs at their very base) but they are not partially molten until they almost reach the Earth's surface. Basalts played no role at all in core formation. Timing of magma ocean crystallisation has been revised since Abe 1997 down to a few Myr only (Lebrun 2013, Hamano 2013), the latter using 0.1 Pa.s for the viscosity of ultra-basic magma ocean.

Previous literature on the properties of magmas at high pressures:

The paper is not quite fair to the theoretical literature. Although most important papers are cited besides Dufils 2017 on the viscosity of basaltic melts at high pressures from MD calculations, such is not the case of their results. Coordination changes have been reported by the papers cited, including the exact pressure at which they occur. Detailed fraction of four-, five- and six-coordinated Al and Si have also been reported as a function of pressure unlike stated I. 118-119. Experimentally, papers based on x-ray diffraction do not provide such detail but do provide averaged coordination numbers (which was done well above 15 GPa by a few papers, cited here although not always where they should). However Petitgirard 2018 did report this from x-ray Raman spectroscopy on high pressure glasses.

Other comments:

Choice of composition: the authors explain that the very high Ca abundance of their composition is the consequence of the simplification (no Fe). Usually, petrologists compensate Fe for Mg, not Ca.

Indeed, as pointed out in the paper, Ca is larger and modifies melt's properties significantly such as increasing its viscosity.

But more importantly, the authors should explain why their results obtained on basalt may be transferred to a magma ocean. That is possible for equations of state with minor corrections, but is more difficult for viscosity which strongly depends on the SiO₂ content.

Melting curve of basalt:

At 2200 K (the lowest of the 2 isotherms investigated here) basalt solidifies at 14 GPa (Hirose 1999). Combination of experimental and theoretical error bars could explain that basalt is still molten in the present work at 2200 K however that can only hold up to 20-22 GPa, after which the melting curve steepens. This could explain the decreased diffusivities (and consequent viscosity increase) reported here. Even at 3000 K, the highest isotherm, basalt solidifies at 50 GPa (Hirose 1999, Gopal 2014) however the authors do not report viscosity below 40 GPa along the 3000 K isotherm.

My question is therefore: may these calculations still have a meaning if done below the melting curve?

Reviewer #2 (Remarks to the Author):

The authors performed ab initio molecular dynamics simulations to investigate the structure and ionic dynamics of a basaltic melt model, Ca₁₁Mg₇Al₈Si₂₂O₇₄, in a pressure range of 0-82GPa at 2,200K and 3,000K. The authors claimed the ionic and weak Si-O and Al-O bonds at high pressure leads to an anomalous trend change in transport properties near 50 GPa, which correlates well with the viscosity decrease in the lower mantle. This is an interesting observation but obviously the mantle is solid and does not have basaltic composition. Besides, the mantle has iron undergoing a spin state change, which the authors cite (not properly though. See below). The support for the melt origin of ULVZ is more relevant. Obviously, the variation of viscosity with melt composition would be most important in this case, especially the presence of iron and hydrogen (also carbon), which the authors mention and acknowledge should be investigated. The importance of these melt properties to understanding the properties of the asthenosphere is also relevant, as discussed on page 11. A comparison of this basaltic melt viscosity with the mantle viscosity structure is a bit of a stretch. Nevertheless, the correlation seems to be real and it might simply be related to the behavior of Si-O and Al-O bond strengths and coordination through the structure. It is, therefore, a relevant observation. There also exist some technical issues in the paper that could be improved. I would suggest the authors add more details to clarify these issues.

1) The authors should clarify the definition of CN in the current liquids. It can be based on a sharp bond-length threshold, or integration of the first peak in $g(r)$, or Voronoi tessellation.

2) The authors described bond angle changes in Page 4-5. It would be more informative to show a typical bond angle analysis, instead of the plain description.

3) The reported diffusivity values are quite small compared to a typical liquid simulation. Mean square displacement should be shown to ensure the equilibrium state of the simulation is liquid and disclose the measurement of the diffusivity.

4) The discussion of diffusivity change is a bit confusing and might be improved. The authors claim Mg and Ca move easily because there are open spaces. But the diffusivity data in Fig.3 shows Ca or Mg are usually the slowest elements under most higher pressures at 2,200K.

5) At 60 GPa and 70 GPa, Mg becomes the slowest specie while it is fastest at 50 GPa and at 80 GPa. Why does the trend change so rapidly? I suggest the authors include the confidence interval for the diffusivity data.

6) The description on Page 8 line 175-177 is confusing. Only the 3,000K data shows an anomalous maximum electrical conductivity at 50 GPa.

7) The Green-Kubo relation in Line 321 on Page 14 is not OK. What happens with the $\alpha\beta$ indices?

8) The authors should provide the plot of $\langle P(t)P(0) \rangle$ to ensure the readers that it converges to 0 for viscosity calculations. It's usually very hard to converge pressure in the ab initio MD timescales.

9) How is the confidence interval of the viscosity data determined in Fig. 4?

10) At 2,200K, the change of viscosity is different from the change of diffusivity, while they are quite consistent at 3,000K. Is the Stock-Einstein relation valid under these conditions?

11) The authors cite papers of papers that mention the effect of the iron spin state change on the viscosity in the lower mantle. The first paper that made a case for this phenomenon is Wentzcovitch, R. M. et al. Anomalous compressibility of ferropericlase throughout the iron spin crossover. Proceedings of the National Academy of Sciences of the United States of America 106, 8447-8452, doi: 10.1073/pnas.0812150106 (2009). Please make sure proper credit is given.

12) I am sure the authors can address most of these issues in their reply without difficulty.

Reviewer #3 (Remarks to the Author):

This manuscript presents results from molecular dynamic calculations for a model basalt composition up to 80 GPa at 2200 and 3000K. The study finds that CN and viscosity generally increase with pressure, while diffusivity and electrical conductivity generally decrease with pressure at 2200K. Geophysical implications for the ~1000km viscosity anomaly, the ULVZs, and the magma ocean are briefly addressed.

In general, I think this paper is well written and presented. I am not an expert on MD calculations and so I am going to move forward with my review assuming that the calculations have been performed without issue. The main issue with the ms in its present form is that there are some critical references and data for 3000K pressure points are missing. These should be relatively minor edits to address. Also, the discussion of the ULVZ should be removed and the discussion for the other geophysical implications should be expanded, as this will make the paper of broader interest for the readership of Nature Geoscience.

Comments:

- Figure 1: thin lines should be added to guide the eye for the CN trends, similar to what is shown in Fig. S6.
- LI.101-102: The data observation or reference to the more ionic nature should be clarified. The previous discussion is about bond lengths, CN and angles, but the connection to the changing nature of the bonding is unclear, especially to the general readership of Nature Geosciences.
- LI.129, 164, 170, and others: The references need to be more diligently attended to. There are several sentences in this manuscript that talk about predictions, expectations, etc. but there is no corresponding reference. This needs to be remedied before publication.
- LI. 137 – The data below 38 GPa should be included in the supplemental material.
- LI.157 – the low pressure data for the 3000K are missing? Were the calculations at 3000K only performed from 40-70GPa? This is unclear to me in the reading of the manuscript, perhaps I just

missed it, but in the methods section (ll. 300) it is implied that there is data from 0-82 GPa at 3000K.

- Ll. 161 – also the diffusion coefficient of oxygen and Si increases in figure 3a (and b) in the 50-70 GPa region.
- Ll. 162 – what do the authors mean that the increase is not an artifact of the calculations? The reasoning for this should be discussed at least in the supplemental data.
- Figure 4: Why do diffusivity/EC and viscosity show opposite trends from 20-50 GPa? Why does diffusivity/EC data at 2200 and 3000K show different trends from 40-70 GPa, while viscosity does not? Also, what are the 'R' in the legend of figure 4? It might be good to clarify this in the figure caption.
- Ll. 263 - the discussion of the ULVZ is beyond scope of the work of this paper (~50 GPa higher pressure and 1000K hotter). This paragraph should be removed and the discussion of the 1000km viscosity anomaly, the implications for magma ocean dynamics, and/or the interpretation of geophysical observables at the mantle transition zone should be expanded.

Reviewers' comments:

Reviewer #1 (Remarks to the Author):

This paper reports the properties of a molten basalt analogue up to 80 GPa by means of theoretical calculations. The major finding is that melt viscosity remains low up to the highest P investigated (80 GPa) with a non-monotonous P-evolution. Viscosity slightly decreases up to 20 GPa, then increases only to reach a maximum near 50 GPa (100 mPa.s) and then slowly decreases. These results differ from those obtained previously from theoretical calculations (Karki 2018 and refs therein) that did not report a slow decrease above 50 GPa, with viscosity values about 2-3 times higher than found here. Although experimental data on viscosity only exist up to modest P, experimental values are also 3-4 times higher than found here (i.e. below 10 GPa, cf highest T points in Sakamaki 2013). The decrease of viscosity above 50 GPa is attributed to the Si-O and Al-O bonds become ionic, explaining weakening of the melt structure and reduced viscosity. The most straightforward implication of such low viscosity is that the magma ocean must have cooled faster than previously thought. There are no molten basalts in the present day deep mantle except in ULVZs. Hence implications for the reported viscosity decrease in the solid lower mantle are quite stretched. There are a number of mistakes in the literature and geological background (see below), however that can be corrected. I am more concerned by the mismatch between the present results on diffusivity and viscosity and previous theoretical and experimental results, while structural and density data are in general agreement with previous works. Such mismatch, especially at modest P at which experiments can be done and are confirmed by other theoretical works, cast doubts on the new finding i.e. a decrease of viscosity at lower mantle conditions. For these reasons, I cannot recommend the paper for publication. Geological background:

I do not think that it is appropriate to state climate change as being governed by the properties of magmas at depth. This induces the idea of a control from the deep Earth on climate change, which is inaccurate and dangerous for a scientist to state. Super-eruptions (e.g. traps) did influence climate in the past, but there is no such effect since CO₂ increase in the atmosphere from human-related activities.

Reply: The calculation of transport properties requires a large model system and very long simulations. Even in the recent past, ab initio calculations could only be performed on a small system (usually < 100 atoms). In this study we expand significantly the number of atoms and carefully monitor the convergence of the calculated properties. An example on the convergence of the calculated viscosity is shown in Fig. S12 in the supplementary information. The First principles results reported here cannot be compared with calculations performed with empirical potentials particularly at very high pressures. Moreover, the trajectories from older ab initio calculations were often shorter, within 10's of ps, and the results may be not reliable. These factors may be potential reasons for the discrepancies. The reviewer has raised concern about discrepancy between our results with that of experiments. This statement is very vague as it does

not mention which theoretical and experimental results. As it is that not much data on the diffusivity and viscosity of basalt exists in the literature, therefore, for decades researchers have resorted to approximate basalt using different models, starting from silicate melts, aluminosilicate melts, model basalt, etc. Thus, it is obvious that there will be differences. However, we would like to point out that this misleading remark by the reviewer is not completely fair to our work as it can severely skew the editorial decision. In fact, our results are in quite close agreement and in the same order of magnitude as that of related works, both theoretical and experimental. Some representative literature that we are obliged to cite here are the following. The viscosities reported in these works at the similar pressures and temperatures are in quite good agreement with our data.

- *Sakamaki, T. et al. Ponded melt at the boundary between the lithosphere and asthenosphere. Nat. Geosci. 6, 1041–1044 (2013).*
- *Dufils, T., Folliet, N., Mantisi, B., Sator, N. & Guillot, B. Properties of magmatic liquids by molecular dynamics simulation: The example of a MORB melt. Chem. Geol. 461, 34–46 (2017).*
- *Cochain, B. et al. Viscosity of mafic magmas at high pressures. Geophys. Res. Lett., 44, 818–826 (2017).*
- *Ghosh, D. B. & Karki, B. B. Transport properties of carbonated silicate melt at high pressure. Sci. Adv. 3, e1701840 (2017).*
- *Dufils, T., Sator, N. & Guillot, B. Properties of planetary silicate melts by molecular dynamics simulation. Chem. Geol. 493, 298–315 (2018).*

We agree with the reviewer and have removed the sentence insinuating climate change being a manifestation of the properties of magma at depths.

Most implications related to the presence of basalts at depth are fine however at places the manuscript implies that there is partial melt in the lower mantle which is not the case except in ULVZs where T are high enough to induce partial melting. Mantle plumes may rise from low-shear wave velocity provinces (that do have some ULVZs at their very base) but they are not partially molten until they almost reach the Earth's surface. Basalts played no role at all in core formation.

Reply: We agree with the reviewer that the lower mantle is predominantly in the solid state and have deleted the entire discussion on the mantle viscosity structure. Instead, we have added new calculations for 120 GPa and 4000 K, directly relevant to the ULVZ at the core-mantle boundary. Our calculated viscosity of 100 ± 10 mPa.s for the model basaltic melt at 120 GPa and 4000 K is almost identical to that of anhydrous MgSiO₃ liquid under the same condition reported by Wan et al. [Wan, J. T. K., Duffy, T. S., Scandolo, S. & Car, R. First-principles study of density, viscosity, and diffusion coefficients of liquid MgSiO₃ at conditions of the Earth's deep mantle. J. Geophys. Res. 112, B03208 (2007)]. This result is important in not only validating our calculations but also suggesting that major constituents such as Ca and Al have minimal effects on transport properties of silicate melts under lower mantle conditions. This result is in direct contradiction from the fact that chemical compositions of silicate melts exert profound impacts

*on their structures and transport properties at low pressures [Ni, H., Hui, H. & Steinle-
neumann, G. Transport properties of silicate melts. Rev. Geophysics **53**, 715–744 (2015)]. In
addition, our calculations at 120 GPa and 4000 K, along with previous calculations [Karki, B.
B. & Stixrude, L. Viscosity of MgSiO₃ Liquid at Earth's mantle conditions: implications for an
early magma ocean. Science **328**, 740–742 (2010); Stixrude, L. & Karki, B. Structure and
freezing of MgSiO₃ liquid in earth's lower mantle. Science **310**, 297–299 (2005); Wan, J. T. K.,
Duffy, T. S., Scandolo, S. & Car, R. First-principles study of density, viscosity, and diffusion
coefficients of liquid MgSiO₃ at conditions of the Earth's deep mantle. J. Geophys. Res. **112**,
B03208 (2007)], provide theoretical basis of extensive melting and generation of basaltic melts
and the formation of superplumes at the core-mantle boundary. Moreover, the viscosity reversal
of the model basaltic melt calculated at pressures of ~50 GPa (Fig. 4) provides the first
tantalizing explanation for the horizontal deflection or stagnation of superplumes at the depth of
~1000 km [French, S. W. & Romanowicz, B. Broad plumes rooted at the base of the Earth's
mantle beneath major hotspots. Nature **525**, 95-99 (2015)].*

Timing of magma ocean crystallisation has been revised since Abe 1997 down to a few Myr only (Lebrun 2013, Hamano 2013), the latter using 0.1 Pa.s for the viscosity of ultra-basic magma ocean.

*Reply: We have modified our discussion on the timescales of magma oceans accordingly. Nevertheless, we emphasize, as noted by the reviewer as well, that a viscosity value of 0.1 Pa.s adopted by Hamano et al. [Hamano, K., Abe, Y. & Genda, H. Emergence of two types of terrestrial planet on solidification of magma ocean. Nature **497**, 607–611 (2013)] was assumed for ultra-basic magma oceans in their radiative-convection equilibrium calculations at 3000 K. However, previous calculations of anhydrous MgSiO₃ liquids at 3000 K [Karki, B. B. & Stixrude, L. Viscosity of MgSiO₃ Liquid at Earth's mantle conditions: implications for an early magma ocean. Science **328**, 740–742 (2010)] suggested that this low viscosity is valid only at pressures below 40 GPa. On the other hand, our calculations with the new reversed trend at ~50-80 GPa yield viscosity values of ~0.1 Pa s for the model basaltic melt under most lower mantle conditions, hence providing further support for the short timescales of magma oceans at a few million years.*

Previous literature on the properties of magmas at high pressures:

The paper is not quite fair to the theoretical literature. Although most important papers are cited besides Dufils 2017 on the viscosity of basaltic melts at high pressures from MD calculations, such is not the case of their results. Coordination changes have been reported by the papers cited, including the exact pressure at which they occur. Detailed fraction of four-, five- and six-coordinated Al and Si have also been reported as a function of pressure unlike stated l. 118-119. Experimentally, papers based on x-ray diffraction do not provide such detail but do provide averaged coordination numbers (which was done well above 15 GPa by a few papers, cited here

although not always where they should). However Petitgirard 2018 did report this from x-ray Raman spectroscopy on high pressure glasses.

Reply: We have now made the citations more robust and have included the ones pointed out by the reviewer in the proper places including clarifying the statements. In Figs. 1b and 1d, of the manuscript, we have even compared the average coordination numbers of Si and Al respectively with those reported in literature, both below and above 15 GPa. At 0 GPa and 2200 K, the average Si-O coordination is calculated to be 4.09 in our work; this is comparable to the value of ~4.06 reported in the theoretical study by Bajgain et al. [Bajgain, S., Ghosh, D. B. & Karki, B. B. Structure and density of basaltic melts at mantle conditions from first-principles simulations. Nat. Commun. 6, 8578 (2015)]. Furthermore, it should be noted that the calculated coordination numbers are sensitive to the choice of the Si-O cutoff radius; in this work we have chosen the cutoff radius to be the first minimum after the first nearest neighbor peak. In an experiment cum theoretical work by Petitgirard et al. on pure silica melts [Petitgirard, S. et al. Magma properties at deep Earth's conditions from electronic structure of silica. Geophys. Perspect. Lett. 9, 32–37 (2019)], the average Si-O coordination is ~4.0 at 0 GPa and 3000K. Our results are also in good agreement at slightly higher pressures with that observed by Sanloup et al. for basalt [Sanloup, C. et al. Structural change in molten basalt at deep mantle conditions. Nature 503, 104–107 (2013)]. While at 18 GPa and 2200 K, the average Si-O coordination we calculated is 4.41, Sanloup et al. obtained this to be 4.5 at 15 GPa and 2273 K. Even the fractional presence of 4,5 and 6 fold coordination of Si with O are in very good agreement with that shown by Petitgirard et al. [Petitgirard, S. et al. Magma properties at deep Earth's conditions from electronic structure of silica. Geophys. Perspect. Lett. 9, 32–37 (2019)]. The trend that there is a steep drop in the four fold coordination to give way to five and six fold coordination of Si in the vicinity of 20 GPa is very clear from our work too.

Other comments:

Choice of composition: the authors explain that the very high Ca abundance of their composition is the consequence of the simplification (no Fe). Usually, petrologists compensate Fe for Mg, not Ca. Indeed, as pointed out in the paper, Ca is larger and modifies melt's properties significantly such as increasing its viscosity. But more importantly, the authors should explain why their results obtained on basalt may be transferred to a magma ocean. That is possible for equations of state with minor corrections, but is more difficult for viscosity which strongly depends on the SiO₂ content. Melting curve of basalt: At 2200 K (the lowest of the 2 isotherms investigated here) basalt solidifies at 14 GPa (Hirose 1999). Combination of experimental and theoretical error bars could explain that basalt is still molten in the present work at 2200 K however that can only hold up to 20-22 GPa, after which the melting curve steepens. This could explain the decreased diffusivities (and consequent viscosity increase) reported here. Even at 3000 K, the highest isotherm, basalt solidifies at 50 GPa (Hirose 1999, Gopal 2014) however the authors do not report viscosity below 40 GPa along the 3000 K isotherm. My question is therefore: may these calculations still have a meaning if done below the melting curve?

*Reply: Our choice of the model basaltic composition is intended to investigate and highlight the effects of Ca and Al on the structures and transport properties of silicate melts under lower mantle conditions, in comparison with previous calculations on anhydrous MgSiO₃ liquids. In particular, our inclusion of Ca in the model basaltic melt is justified by the possible presence of the perovskite-structure CaSiO₃ as the third most abundant mineral in the lower mantle. Our calculations show that the investigated system at 2200-3000 K remains molten at pressures up to 80 GPa. Computations of this type adopted in this study are extremely expensive. The pressure range of 40-70 GPa at 3000 K is chosen to validate the viscosity reversal observed at 2200 K. Also, a temperature of 3000 K is unlikely in the upper mantle or the transition zone, hence no calculations at 3000 K and <40 GPa have been performed. Likewise, the highest temperature of 4000 K is possible only at the core-mantle boundary. Therefore, only calculations were made for the model basaltic melts at 4000 K and 120 GPa and are intended for comparison with anhydrous MgSiO₃ liquids under the same condition [Wan, J. T. K., Duffy, T. S., Scandolo, S. & Car, R. First-principles study of density, viscosity, and diffusion coefficients of liquid MgSiO₃ at conditions of the Earth's deep mantle. J. Geophys. Res. **112**, B03208 (2007)].*

Reviewer #2 (Remarks to the Author):

The authors performed ab initio molecular dynamics simulations to investigate the structure and ionic dynamics of a basaltic melt model, Ca₁₁Mg₇Al₈Si₂₂O₇₄, in a pressure range of 0-82GPa at 2,200K and 3,000K. The authors claimed the ionic and weak Si-O and Al-O bonds at high pressure leads to an anomalous trend change in transport properties near 50 GPa, which correlates well with the viscosity decrease in the lower mantle. This is an interesting observation but obviously the mantle is solid and does not have basaltic composition. Besides, the mantle has iron undergoing a spin state change, which the authors cite (not properly though. See below). The support for the melt origin of ULVZ is more relevant. Obviously, the variation of viscosity with melt composition would be most important in this case, especially the presence of iron and hydrogen (also carbon), which the authors mention and acknowledge should be investigated. The importance of these melt properties to understanding the properties of the asthenosphere is also relevant, as discussed on page 11. A comparison of this basaltic melt viscosity with the mantle viscosity structure is a bit of a stretch. Nevertheless, the correlation seems to be real and it might simply be related to the behavior of Si-O and Al-O bond strengths and coordination through the structure. It is, therefore, a relevant observation. There also exist some technical issues in the paper that could be improved. I would suggest the authors add more details to clarify these issues.

1) The authors should clarify the definition of CN in the current liquids. It can be based on a sharp bond-length threshold, or integration of the first peak in g(r), or Voronoi tessellation.

Reply: We thank the reviewer for pointing this out and have included the description in the methods section of the manuscript also. The CN has been calculated by integration of the first nearest neighbour peak of the radial distribution function, $g(r)$, up to the first minimum.

2) The authors described bond angle changes in Page 4-5. It would be more informative to show a typical bond angle analysis, instead of the plain description.

Reply: On request from the reviewer, we have included bond angle distribution plots for O-Si-O and O-Al-O for different pressure points at 2200 K, in the supplementary section. The $\sim 90^\circ$ is the octahedral O-Si-O and the $\sim 170^\circ$ is the axial (linear) O-Si-O showing that the local environment is almost octahedral. The O-Al-O becomes 6 coordinated much faster than O-Si-O. This is also reflected in the coordination number analysis. This has also been written in the manuscript now and we hope will assist in more lucid reading of the manuscript by the readers.

3) The reported diffusivity values are quite small compared to a typical liquid simulation. Mean square displacement should be shown to ensure the equilibrium state of the simulation is liquid and disclose the measurement of the diffusivity.

*Reply: We have now shown the mean square displacement for the different species at 0 GPa and 2200 K as a representative. However, we respectfully disagree with the reviewer regarding to the magnitudes of the diffusion coefficients. The diffusivity values are absolutely in the proper order of magnitude. In literature these values vary from 10^{-12} to 10^{-9} cm^2/s and all our values fall in this range. For instance, at ambient pressure and 2200 K, the self-diffusivities from our calculations for Ca, Mg, Al, Si and O are 2.35×10^{-10} , 3.65×10^{-10} , 1.93×10^{-10} , 1.33×10^{-10} and 2.0×10^{-10} m^2/s , respectively. The same at 50 GPa and 3300 K are 1.54×10^{-10} , 1.7×10^{-10} , 2.67×10^{-10} , 2.01×10^{-10} and 2.89×10^{-10} m^2/s . According Ghosh and Karki [Ghosh, D. B. & Karki, B. B. Transport properties of carbonated silicate melt at high pressure, *Sci. Adv.* **3**, e1701840 (2017)] for MgSiO_3 , the approximate self-diffusivities for Mg, Si and O are 1.1×10^{-9} , 2.0×10^{-10} and 6.0×10^{-10} m^2/s at 0 GPa and 2200 K. The same at 50 GPa and 3000 K are 6.0×10^{-10} , 4.0×10^{-10} and 6.0×10^{-10} m^2/s . According to Dufils et al. [Dufils, T., Folliet, N., Mantsi, B., Sator, N. & Guillot, B. Properties of magmatic liquids by molecular dynamics simulation: The example of a MORB melt. *Chem. Geol.* **461**, 34–46 (2017)], for mid-ocean ridge basalt, the calculated diffusion coefficients at 2073 K are approximately 6.0×10^{-10} , 6.0×10^{-10} , 2.0×10^{-10} , 0.35×10^{-10} and 0.7×10^{-10} m^2/s for Ca, Mg, Al, Si and O, respectively. Thus, we can see that all our numbers are in pretty good agreement with that of other theoretical works.*

4) The discussion of diffusivity change is a bit confusing and might be improved. The authors claim Mg and Ca move easily because there are open spaces. But the diffusivity data in Fig.3 shows Ca or Mg are usually the slowest elements under most higher pressures at 2,200K.

Reply: At 60 and 70 GPa, the diffusivities have been recalculated and definitely more plausible magnitudes have been reported now. We apologize for the error.

5) At 60 GPa and 70 GPa, Mg becomes the slowest specie while it is fastest at 50 GPa and at 80 GPa. Why does the trend change so rapidly? I suggest the authors include the confidence interval for the diffusivity data.

Reply: At 60 and 70 GPa, the diffusivities have been recalculated and definitely more plausible magnitudes have been reported now. We apologize for the error. It is very difficult to show confidence levels in log plots. So we have written the confidence levels in the manuscript.

6) The description on Page 8 line 175-177 is confusing. Only the 3,000K data shows an anomalous maximum electrical conductivity at 50 GPa.

Reply: We have added a line in the relevant section that for 2200 K, an anomalous maximum is also seen at 60 GPa. Hence, anomalous maxima are seen for both 2200 and 3000 K. This should remove any confusion.

7) The Green-Kubo relation in Line 321 on Page 14 is not OK. What happens with the $\alpha\beta$ indices?

Reply: The Green-Kubo equation had a factor of 3 missing in the denominator. It was a typo and has now been corrected. We thank the referee for pointing this out. The meaning of $\alpha\beta$ have also been explained in the revision. The indices refer to the three off-diagonal elements of the stress tensor, namely, xy , yz , and zx components. Therefore, the theoretical background and reproducibility of our calculations has been ensured.

8) The authors should provide the plot of to ensure the readers that it converges to 0 for viscosity calculations. It's usually very hard to converge pressure in the ab initio MD timescales.

Reply: Within the time scale of ab initio MD, sometimes it is indeed difficult to converge the stress autocorrelation function (SACF) to 0. Hence, a plot showing the decay of the SACF and convergence to zero has been added in the supplementary section (Fig. S11). This valuable suggestion definitely makes the viscosity analysis more convincing.

9) How is the confidence interval of the viscosity data determined in Fig. 4?

Reply: We divided the MD trajectory into several time segments with different time origins to compute the correlation functions while making sure that convergence had been achieved in each segment as described above. The error was the standard deviation from the mean value. We have now explained this in the methods section.

10) At 2,200K, the change of viscosity is different from the change of diffusivity, while they are quite consistent at 3,000K. Is the Stock-Einstein relation valid under these conditions?

*Reply: The Stokes-Einstein relation between particle diffusivity and fluid viscosity works well for comparatively simpler liquids such as metals [Poirier, J. P., Transport properties of liquid metals and viscosity of the Earth's core. Geophys. J. **92**, 99-105, (1988); Dobson, D. P., et al., In situ measurement of viscosity of liquids in the Fe-FeS system at high pressures and temperatures. Am Mineral **85**, 1838-1842 (2000)]. However, there have been reports about discrepancies for fluids of more complex composition implying that the diffusion involves mechanisms that cannot be explained by such a simple model [Rev. Mineral. Geochem. **72**, 971-996 (2010)]. In the Stokes-Einstein relation, the effective hydrodynamic radius of the atoms has to be either taken as an average of the different species involved or that of the atoms which contribute majorly to the diffusivity. This can give rise to significant errors and therefore the Green-Kubo method of finding out the viscosity from MD is definitely more reliable.*

11) The authors cite papers of papers that mention the effect of the iron spin state change on the viscosity in the lower mantle. The first paper that made a case for this phenomenon is Wentzcovitch, R. M. et al. Anomalous compressibility of ferropericlase throughout the iron spin crossover. Proceedings of the National Academy of Sciences of the United States of America **106**, 8447-8452, doi:10.1073/pnas.0812150106 (2009). Please make sure proper credit is given.

Reply: This is a justified claim. In the revised manuscript, as requested by this reviewer, we have removed the major discussion on the ULVZ and the geological implications of our results on it.

Reviewer #3 (Remarks to the Author):

This manuscript presents results from molecular dynamic calculations for a model basalt composition up to 80 GPa at 2200 and 3000K. The study finds that CN and viscosity generally increase with pressure, while diffusivity and electrical conductivity generally decrease with pressure at 2200K. Geophysical implications for the ~1000km viscosity anomaly, the ULVZs, and the magma ocean are briefly addressed.

In general, I think this paper is well written and presented. I am not an expert on MD calculations and so I am going to move forward with my review assuming that the calculations have been performed without issue. The main issue with the ms in its present form is that there are some critical references and data for 3000K pressure points are missing. These should be relatively minor edits to address. Also, the discussion of the ULVZ should be removed and the discussion for the other geophysical implications should be expanded, as this will make the paper of broader interest for the readership of Nature Geoscience.

Comments:

- Figure 1: thin lines should be added to guide the eye for the CN trends, similar to what is shown in Fig. S6.

Reply: We have now joined the points in Fig. 1 for better aid to the eye.

- Ll.101-102: The data observation or reference to the more ionic nature should be clarified. The previous discussion is about bond lengths, CN and angles, but the connection to the changing nature of the bonding is unclear, especially to the general readership of Nature Geosciences.

Reply: When the coordination changes from four fold to six fold, bond lengths increase to accommodate more atoms in the polyhedron. Therefore, the bonds become weaker going through an eventual transition from covalent (strong) to ionic (weak). We have added a line on this in the manuscript too.

- Ll.129, 164, 170, and others: The references need to be more diligently attended to. There are several sentences in this manuscript that talk about predictions, expectations, etc. but there is no corresponding reference. This needs to be remedied before publication.

Reply: We agree with the reviewer and have now cited the appropriate references accordingly.

- Ll. 137 – The data below 38 GPa should be included in the supplemental material.

Reply: We have kept this for better representation to show the low pressure tendencies as well. This is important to show the change in the structural properties that lead to the change in the transport properties.

- Ll.157 – the low pressure data for the 3000K are missing? Were the calculations at 3000K only performed from 40-70GPa? This is unclear to me in the reading of the manuscript, perhaps I just missed it, but in the methods section (ll. 300) it is implied that there is data from 0-82 GPa at 3000K.

Reply: The aim for performing calculations at 3000K was to verify the anomaly observed in the viscosity at 2200K. Calculations of viscosity are computationally very demanding. Therefore, we restricted calculations to the relevant pressure range comparable to the 2200K. Moreover, basaltic melt does not exist in the nature at high temperature and low pressure. No relevant experiment data is available or comparison. To corroborate the diffusion and viscosity results obtained at 2200 K in the pressure range 40-70 GPa, we calculated the same at 3000 K in the 40-70 GPa range too. We have now clarified this in the methods section of the manuscript.

- Ll. 161 – also the diffusion coefficient of oxygen and Si increases in figure 3a (and b) in the 50-70 GPa region.

Reply: Indeed, and we have now corrected this.

• Ll. 162 – what do the authors mean that the increase is not an artifact of the calculations? The reasoning for this should be discussed at least in the supplemental data.

Reply: We have clarified this statement in the manuscript. We meant that it is not a one-time calculation error rather has been reproduced under different conditions too.

• Figure 4: Why do diffusivity/EC and viscosity show opposite trends from 20-50 GPa? Why does diffusivity/EC data at 2200 and 3000K show different trends from 40-70 GPa, while viscosity does not? Also, what are the ‘R’ in the legend of figure 4? It might be good to clarify this in the figure caption.

Reply: Between 20 and 50 GPa, the diffusivity/electrical conductivity decreases while viscosity increases which is obvious due to the reduced mobility of the ions.

Between 40 and 70 GPa, the reviewer has pointed that the trends in the diffusivities are different at 2200 and 3000 K while the trend is same for viscosity. We do not totally agree with this comment. All the species except for Mg show very similar behavior, i.e. Ca, Al, Si and O have sudden kink at 50 or 60 GPa at both 2200 and 3000 K while Mg does not. Moreover, at 70 GPa, only Mg has a slightly increased value of diffusivity. Within the confidence level now reported in the manuscript, this change is minute. Lastly, Ca and Mg are network modifiers while Si and Al are network makers. Network makers are dominant in determining the viscosity. Therefore, the almost same trends of Si, Al and O at 2200 and 3000 K, render the same viscosity trends at both the temperatures.

The R is density scaling as defined in Ref. 36 as $R_{\rho,s} = d \ln \rho / d \ln V_s$ as defined in Ref. 36 of the original manuscript. However, in this revised version, we have removed this discussion.

• Ll. 263 - the discussion of the ULVZ is beyond scope of the work of this paper (~50 GPa higher pressure and 1000K hotter). This paragraph should be removed and the discussion of the 1000km viscosity anomaly, the implications for magma ocean dynamics, and/or the interpretation of geophysical observables at the mantle transition zone should be expanded.

Reply: We have now added the results and a discussion on an additional calculation performed at 120 GPa and 4000 K which is directly relevant to the discussion of the ULVZ. Also, we have highlighted the viscosity anomaly at ~50 GPa providing the first tantalizing explanation for the horizontal deflections of superplumes from the core-mantle boundary.

REVIEWERS' COMMENTS:

Reviewer #2 (Remarks to the Author):

The revised manuscript addressed all my comments satisfactorily. I also see that the authors went a long way to address the concerns of other referees. From my perspective, this is a well presented and thoughtful piece of research reporting a curious drop in viscosity of a basaltic melt above 50 GPa. Geophysical implications are not always obvious and can be argued about, but the results seem credible, and that is more important to me. The latter will stand, while the former may not, depending on progress in understanding in the field. I believe the paper is suitable for publication.

Reviewer #3 (Remarks to the Author):

This study investigates the effect of pressure and temperature on the transport properties of basaltic melt model composition $\text{Ca}_{11}\text{Mg}_7\text{Al}_8\text{Si}_{22}\text{O}_{74}$, at pressures from 0-120 GPa and in the temperature range of 2200- 4000K (not all pressures and temperatures combined).

The added supplemental figures (i.e. S1b, S1d, S2b, S1d) have improved the comparison to previous data and show many of the same general trends, which was a concern of all the previous reviewers. The authors also found a calculation error that has been corrected (e.g. Reviewer 2 point 4). Also, the focus of the discussion has been shifted and now focuses on plumes and the ULVZ. The applicability of the data to

Generally, I feel that the requested issue that were brought up by the previous reviewers have been addressed satisfactorily. I have two outstanding concerns prior to publication. They are given below. There are a few minor typos (e.g. ll. 193), I would consider a final edit to clean up the ms.

Comments:

1) Focusing on the ULVZ and not having Fe present in the melt does create some questions as the ULVZs are thought to possibly be Fe rich – they are sitting atop the core – a giant vat of Fe (and other elements). More over while the 120 GPa data point is at significantly higher pressure than the previous versions data – which had a maximum pressure of 82 GPa – the CMB is at 135 GPa and the ULVZs are only 10's on km high – meaning that the data calculated here for 120 GPa is still 100's of kms (approaching 500km) too shallow. Considering these two issues (lack of Fe and low pressure), I would consider still removing this discussion and focusing on the plume story. In fact, I would consider removing the 4000K, 120 GPa data point all together and save it for a followup study or something. As shown in Figure 4, there is every little that can be said about viscosity at the CMB as there is no trend present (i.e. only one point). There could be a follow up section in the discussion like "If this trend continues with depth and temperature, there may be implications for the ULVZ by" but anything more is really too speculative with the data present.

2) The added figures in this version of the manuscript are nice. I would suggest adding the diffusivity and viscosity data of others (both calculations and experiments) to Figure 3 and 4. I.e. refs 19-21, 2-4, etc. At a minimum this type of plot or table should be included in the supplemental material. As long as this data is generally consistent, I would think that this manuscript is ready for publication.

Reviewer #2 (Remarks to the Author):

The revised manuscript addressed all my comments satisfactorily. I also see that the authors went a long way to address the concerns of other referees. From my perspective, this is a well presented and thoughtful piece of research reporting a curious drop in viscosity of a basaltic melt above 50 GPa. Geophysical implications are not always obvious and can be argued about, but the results seem credible, and that is more important to me. The latter will stand, while the former may not, depending on progress in understanding in the field. I believe the paper is suitable for publication.

We thank the reviewer for the positive evaluation and the recommendation for acceptance for publication.

Reviewer #3 (Remarks to the Author):

This study investigates the effect of pressure and temperature on the transport properties of basaltic melt model composition $\text{Ca}_{11}\text{Mg}_7\text{Al}_8\text{Si}_{22}\text{O}_{74}$, at pressures from 0-120 GPa and in the temperature range of 2200- 4000K (not all pressures and temperatures combined).

The added supplemental figures (i.e. S1b, S1d, S2b, S1d) have improved the comparison to previous data and show many of the same general trends, which was a concern of all the previous reviewers. The authors also found a calculation error that has been corrected (e.g. Reviewer 2 point 4). Also, the focus of the discussion has been shifted and now focuses on plumes and the ULVZ.

Generally, I feel that the requested issue that were brought up by the previous reviewers have been addressed satisfactorily. I have two outstanding concerns prior to publication. They are given below. There are a few minor typos (e.g. ll. 193), I would consider a final edit to clean up the ms.

We thank the reviewer for the critical comments and further suggestions.

Comments:

1) Focusing on the ULVZ and not having Fe present in the melt does create some questions as the ULVZs are thought to possibly be Fe rich – they are sitting atop the core – a giant vat of Fe (and other elements). More over while the 120 GPa data point is at significantly higher pressure than the previous versions data – which had a maximum pressure of 82 GPa – the CMB is at 135 GPa and the ULVZs are only 10's on km high – meaning that the data calculated here for 120 GPa is still 100's of kms (approaching 500km) too shallow. Considering these two issues (lack of Fe and low pressure), I would consider still removing this discussion and focusing on the plume story. In fact, I would consider removing the 4000K, 120 GPa data point all together and save it for a follow up study or something. As shown in Figure 4, there is every little that can be

said about viscosity at the CMB as there is no trend present (i.e. only one point). There could be a follow up section in the discussion like “If this trend continues with depth and temperature, there may be implications for the ULVZ by” but anything more is really too speculative with the data present.

Reply: We agree with the assessments. One of the reasons not to include other elements, particularly Fe in the model is due to increased complexity of the calculations. At low pressure Fe^{2+} is probably in the high spin state. Whilst at higher pressure, a high spin to low spin transition may occur. This is an interesting problem in its own right. However, it is well-known that the electronic state of Fe ions cannot be described adequately with the PBE functional without the consideration of correlation effect. The empirical Hubbard correction may be a viable alternative, but it is also known that the on-site repulsion parameter U is pressure dependent. Therefore, inclusion of Fe ions in the model is not a trivial task. This topic, perhaps, will be an interesting subject for future studies. Apart from correcting the typos as a concluding remark, we added the following lines at the beginning of the discussion/implication section explaining the issue raised by the reviewer.

“In comparison with previous calculations in simple systems such as SiO_2 , $MgSiO_3$ and $CaAl_2Si_2O_9$ ¹⁹⁻²¹, the multiple-component system of $Ca_{11}Mg_7Al_9Si_{22}O_{74}$ investigated in the present study represents a significant step towards natural basaltic compositions but is still missing several components such as Fe and H. However, inclusion of these additional components, especially Fe, requires considerations of oxidation-reduction states and magnetic contributions with possible spin transitions, which are impossible for current computation capabilities. Our calculations for the more normal mantle temperature at 2200 K predict a complete solidification of the model basaltic system above 82 GPa is consistent with the fact that the lower mantle is dominantly in the solid state.”

We also agree that we should remove the discussion on the calculations at 4000 K. In the revision all results and discussion related to 120 GPa and 4000K have been deleted. Removal of the results in no way affect the main conclusion of this study. Furthermore, as suggested by the reviewer, we also added the following to paragraph 2 p.12: “If the predicted trend (Fig. 4) continues with depth and temperature, there may be implications for the ULVZ with significantly lower viscosity than previously suggested.”

2) The added figures in this version of the manuscript are nice. I would suggest adding the diffusivity and viscosity data of others (both calculations and experiments) to Figure 3 and 4. I.e. refs 19-21, 2-4, etc. At a minimum this type of plot or table should be included in the supplemental material. As long as this data is generally consistent, I would think that this manuscript is ready for publication.

Reply: We thank the reviewer for the useful suggestion. We have added the experimental and previous theoretical diffusion and viscosity data and they can be found in the Supplementary Information in Supplementary Figure 9 and Supplementary Figure 10 respectively.